# Integrated Multi-Omics Analysis Reveals Mountain-Cultivated Ginseng Ameliorates Cold-Stimulated Steroid-Resistant Asthma by Regulating Interactions among Microbiota, Genes, and Metabolites

**DOI:** 10.3390/ijms25169110

**Published:** 2024-08-22

**Authors:** Daohao Tang, Chao Wang, Hanlin Liu, Junzhe Wu, Luying Tan, Sihan Liu, Haoming Lv, Cuizhu Wang, Fang Wang, Jinping Liu

**Affiliations:** 1School of Pharmaceutical Sciences, Jilin University, Changchun 130021, China; tangdh22@mails.jlu.edu.cn (D.T.); hanji23@mails.jlu.edu.cn (H.L.); wujz23@mails.jlu.edu.cn (J.W.); tanly23@mails.jlu.edu.cn (L.T.); lvhm2822@mails.jlu.edu.cn (H.L.); wangcuizhu@jlu.edu.cn (C.W.); 2College of Basic Medical Sciences, Jilin University, Changchun 130021, China; wangc19@mails.jlu.edu.cn; 3College of Animal Science and Technology, Jilin Agricultural University, Changchun 130118, China; liusihan040306@126.com

**Keywords:** mountain-cultivated ginseng, steroid-resistant asthma, cold-stimulated, multi-omics, synergistic effects

## Abstract

Steroid-resistant asthma (SRA), resisting glucocorticoids such as dexamethasone (DEX), is a bottleneck in the treatment of asthma. It is characterized by a predominantly neutrophilic inflammatory subtype and is prone to developing into severe refractory asthma and fatal asthma. Currently, there is a lack of universally effective treatments for SRA. Moreover, since cold stimulation does increase the risk of asthma development and exacerbate asthma symptoms, the treatment of cold-stimulated SRA (CSRA) will face greater challenges. To find effective new methods to ameliorate CSRA, this study established a CSRA mouse model of allergic airway inflammation mimicking human asthma for the first time and evaluated the alleviating effects of 80% ethanol extract of mountain-cultivated ginseng (MCG) based on multi-omics analysis. The results indicate that cold stimulation indeed exacerbated the SRA-related symptoms in mice; the DEX individual treatment did not show a satisfactory effect; while the combination treatment of DEX and MCG could dose-dependently significantly enhance the lung function; reduce neutrophil aggregation; decrease the levels of LPS, IFN-γ, IL-1β, CXCL8, and IL-17; increase the level of IL-10; alleviate the inflammatory infiltration; and decrease the mucus secretion and the expression of MUC5AC. Moreover, the combination of DEX and high-dose (200 mg/kg) MCG could significantly increase the levels of tight junction proteins (TJs), regulate the disordered intestinal flora, increase the content of short-chain fatty acids (SCFAs), and regulate the abnormal gene profile and metabolic profile. Multi-omics integrated analysis showed that 7 gut microbes, 34 genes, 6 metabolites, and the involved 15 metabolic/signaling pathways were closely related to the pharmacological effects of combination therapy. In conclusion, integrated multi-omics profiling highlighted the benefits of MCG for CSRA mice by modulating the interactions of microbiota, genes, and metabolites. MCG shows great potential as a functional food in the adjuvant treatment of CSRA.

## 1. Introduction

Asthma is a heterogeneous disease characterized by the chronic inflammation of the airways involving a variety of cells (eosinophils, neutrophils, etc.) and cellular components. It is projected that, by 2025, approximately 400 million individuals globally will be affected by asthma [1]. Clinically, glucocorticoids are mainly applied to control airway inflammation, but 5~20% of patients with massive infiltration of neutrophils in the airways are insensitive to glucocorticoid therapy, which is defined as steroid-resistant asthma (SRA) [2]. The acute exacerbation risk associated with this subtype of asthma is notably elevated, consuming more than 50% of medical resources and resulting in a mortality rate of more than 30% [3,4]. Nontypeable Haemophilus influenzae (NTHi), a pod-less Gram-negative bacteria, is normally present in the upper respiratory tract of healthy adults. But in patients with SRA, it is predominantly colonized in the lower respiratory tract and is closely associated with steroid resistance [5]. In a mouse model of allergic asthma, NTHi infection converted Th2-mediated eosinophilic inflammation into Th17-driven neutrophilic inflammation, leading to severe steroid resistance [6].

Cold stimulation, as a “trigger point”, could aggravate asthma by the infiltration of inflammatory factors, increasing mucus secretion, and impairment of airway epithelial integrity [7,8]. An epidemiological study in Finland has shown that cold weather (15.8~17.7 °C) could exacerbate respiratory symptoms in asthma patients [9]. The time-stratified case-crossover study in Beijing (China) indicated that there was an association between ambient temperature and adult asthma hospitalizations, and most of the burden was attributable to moderate cold (−6.5~22 °C) [10]. The time-series study in Hong Kong for the short-term association between asthma and hospitalizations indicated that adult admissions were sensitive to temperature in the cold season with the cumulative relative risk for lags 0~3 days for 12 °C vs. 25 °C being 1.33 [11]. In asthmatic mice, cold stimulation at 10 °C was also reported to exacerbate airway inflammation, impair lung function, and induce mucus hypersecretion [12]. Moreover, numerous clinical experiments in Russia have demonstrated that cold air leads to the development of airway hyperresponsiveness in 60~80% of asthmatics, which exacerbates the severity of neutrophilic asthma (a form of SRA) [13]. Currently, clinical medications for the treatment of SRA include corticosteroid combinations, biologics, and macrolide antibiotics, but the overall efficacy is less than satisfactory [14,15]. Finding novel approaches including combined medication for treating SRA (especially cold-stimulated SRA) becomes crucial.

Ginseng, a perennial plant of the *Panax* genus, is a traditional Chinese functional food that has been clinically used for over two thousand years worldwide [16]. Due to effectively improving airway inflammation and soothing bronchial smooth muscle, ginseng prescriptions are commonly used in the clinical treatment of asthma [17]. In addition, ginseng also showed significant mitigating effects in a mouse model of asthma, including the improvement in inflammatory infiltration and airway remodeling, and the attenuation of airway hyperresponsiveness [18]. However, the effects of ginseng on SRA have not been reported. Mountain-cultivated ginseng, sown and grown naturally in the wild state of mountains and forests, is typically harvested after 10~20 years and has a similar composition and effects to wild ginseng [19]. Due to the higher quality than garden-cultivated ginseng, it is reported to have a better antioxidant, anti-inflammatory, and immune-enhancing properties [20,21]. Ginsenosides are the main active components of ginseng, with the glucocorticoid-like structure, have been reported to synergistically enhance the therapeutic effects of dexamethasone (DEX) in corticosteroid-dependent diseases, such as systemic lupus erythematosus and alcoholic hepatitis [22,23]. Based on the above, we propose to use MCG in combination with DEX for the treatment of cold-stimulated SRA (CSRA). In the current study, in order to investigate the synergistic effects of MCG on DEX treatment for CSRA in mice, a mouse model of cold-stimulated (10 ± 2 °C) SRA was established for the first time and intervened with a combination of MCG and DEX; the therapeutic benefits on pulmonary function, neutrophil count, inflammatory reaction, mucus secretion, barrier integrity, intestinal flora, gene transcription, and metabolites were all investigated; the “flora–gene–metabolite– pathway” multidimensional network was then established by using the multi-omics integrated analysis to explore the potential mechanism. This study could provide experimental data to elucidate the alleviating effects of MCG and DEX combined therapy on CSRA in mice.

## 2. Results

### 2.1. Chemical Content Analysis of MCG

The total ginsenosides content in the MCG powder is 81.45%. Typical chromatograms of MCG test solutions and standard samples are shown in Figure 1A,B. The chemical contents were ginsenoside Rg1 (10.55%), Re (9.51%), PPD (1.75%), Rg2 (3.12%), Rb1 (18.28%), Rc (4.53%), Rb2 (5.08%), Rd (3.29%), Rg6 (0.27%), F4 (4.71%), Rh4 (3.32%), (20S)-Rg3 (1.64%), PPT (0.59%), Rk1 (0.26%), and Rf (2.27%). The results reveal that the total content of 15 monomeric ginsenosides accounted for 69.20% of MCG powder.

### 2.2. Combination of DEX and MCG Attenuated Lung Function Impairment

The determination results for pulmonary function are shown in Figure 1. With the increase in methacholine concentration, the resistance of inspiratory (RI) values of mice in each group continued to rise, and the dynamic respiratory compliance (Cdyn) values continued to fall (Figure 1C,D). The RI and Cdyn data at the final concentration of 25 mg/mL were selected for statistical analysis (Figure 1E,F). It was shown that the cold stimulation (10 ± 2 °C) had no significant effects on RI and Cdyn in the cold-stimulated control (CSC) group compared with the normal control (NC) group. Compared to the asthma model (AM) group, there was a noticeable reduction in Cdyn and an appreciable elevation in RI in the cold-stimulated asthma model (CAM) group. Compared to the steroid-resistant asthma model (SRAM) group, RI was significantly increased and Cdyn was significantly decreased in the cold-stimulated steroid-resistant asthma model (CSRAM) group. It could be concluded that cold stimulation could significantly aggravate airway hyperresponsiveness in both asthma and SRA mice. As for the mice in the CSRAM group that intervened with DEX, although RI or Cdyn could be decreased or increased, there was no statistically significant difference compared to the CSRAM group. As for the CSRAM + DEX + MCG_L_, CSRAM + DEX + MCG_M_, and CSRAM + DEX + MCG_H_ groups, the RI and Cdyn values were significantly re-regulated in a dose-dependent way compared to the CSRAM group. Additionally, compared to the CSRAM + DEX group, the RI and Cdyn values of the CSRAM + DEX + MCG_M_ and CSRAM + DEX + MCG_H_ groups were all significantly regulated. It was suggested that MCG showed the synergistic effects of DEX on enhancing pulmonary function.

### 2.3. Combination of DEX and MCG Reduced the Proportion and Count of Neutrophils

The neutrophils in bronchoalveolar lavage fluid (BALF) were examined using flow cytometry (NovoCyte, ACEA, San Diego, CA, USA). The representative images for the proportion of neutrophils were presented in Figure 2A, and the counts of neutrophils were listed in Figure 2B. Cold stimulation had no significant effects on the proportion and count of neutrophils in normal mice. However, the proportion and count of neutrophils were elevated in both the CAM and CSRAM groups compared to the AM and SRAM groups; especially, the count of neutrophils in the CSRAM group was significantly increased compared to the SRAM group. It could be concluded that cold stimulation worsens asthma and SRA by increasing neutrophils in mice. Although DEX individual treatment could reduce the number of neutrophils, there was no statistically significant difference compared to the CSRAM group. DEX + MCG combined middle- and high-dose groups to prominently diminish neutrophil counts compared to the CSRAM group, suggesting a pronounced synergistic interaction between MCG and DEX in modulating neutrophil levels in mice.

### 2.4. Combination of DEX and MCG Modulated the Levels of Cytokines

The results of ELISA for each indicator were shown in Figure 3. Compared to the NC group, cold stimulation had no significant effects on the levels of IFN-γ, IL-1β, IL-10, CXCL8, and IL-17 in BALF or the serum of the CSC group. While the levels of these cytokines changed in the CAM group, the differences between the AM group and the CAM group were of no statistical significance. As for the CSRAM group, the levels of IFN-γ in BALF and serum, as well as the levels of IL-1β and IL-17 in BALF, were all significantly increased compared to the SRAM group. The levels of CXCL8 in BALF or serum were obviously increased in the SRAM and CSRAM groups compared to the AM and CAM groups, but the level of CXCL8 was not significantly affected by cold stimulation. Overall, AM and SRAM could be aggravated by cold stimulation by affecting the levels of IFN-γ, IL-1β, and IL-17. Compared to the CSRAM group, the individual prevention of DEX was able to significantly down-regulate the levels of IL-17 (in BALF) and CXCL8 (in serum) and could significantly up-regulate the levels of IL-10 in BALF and serum. This indicates that DEX could re-regulate LPS, IL-17, IL-10, and CXCL8 to a certain extent.

In comparison to the CSRAM group: (1) In the CSRAM + DEX + MCG_H_ group, the levels of IFN-γ, IL-1β, IL-10, and CXCL8 in BALF and serum could be clearly re-regulated; (2) In the CSRAM + DEX + MCG_M_ group, the levels of LPS, IFN-γ, IL-10, and CXCL8 in BALF and serum, and the level of IL-1β in BALF could be re-regulated obviously; (3) In the CSRAM + DEX + MCG_L_ group, the levels of LPS and IL-10 in BALF and serum, the level of IFN-γ in BALF, and the level of CXCL8 in serum could be re-regulated obviously.

More meaningfully, DEX combined with MCG_H_ could enhance the efficacy of DEX statistically on IFN-γ, IL-1β, IL-10, and CXCL8 in BALF or serum; DEX combined with MCG_M_ could enhance the efficacy of DEX statistically on LPS, CXCL8, and IL-10 in BALF or serum, and IL-1β in BALF. This suggests that the co-administration of MCG and DEX exerted a synergistic effect on cytokines in the serum and BALF.

### 2.5. Combination Treatment of DEX and MCG Alleviated Airway Remodeling and Mucus Hypersecretion

H&E staining (Figure 4A) showed that there was no obvious infiltration of inflammatory cells around the perivascular and peribronchial areas in the lung tissues of NC or CSC group mice, while the infiltration of inflammatory cells appeared in the AM group and SRAM group. Compared to the AM group and SRAM group, the CAM group and CSRAM group showed a significantly increased infiltration of eosinophilic or neutrophilic cells and severe airway remodeling (such as significant thickening of the airway wall, narrowing of the lumen, and airway deformation), which further confirmed that cold stimulation worsened asthma and SRA in mice. Compared to the CSRAM group, the individual administration of DEX had no significant effects on airway remodeling, but MCG in combination with DEX was able to dose-dependently reduce inflammatory cell infiltration and morphological changes in the airways.

AB-PAS staining (Figure 4B) showed no significant proliferation of goblet cells and no significant secretion of mucus in the NC and CSC groups, while the AM and SRAM groups showed goblet cell proliferation and mucus secretion. Compared to the AM and SRAM groups, the CAM and CSRAM groups showed a significant increase in goblet cell proliferation and mucus secretion in the airway epithelium, further confirming that cold stimulation exacerbated asthma and SRA in mice. Compared to the CSRAM group, DEX alone did not have a significant inhibitory effect on goblet cell proliferation, while MCG combined with DEX was able to inhibit goblet cell proliferation and reduce mucus secretion, and there was a dose dependency in the combination therapy group.

Immunohistochemical staining (Figure 4C) showed no apparent high expression of MUC5AC in the NC and CSC groups. On the contrary, the AM and SRAM groups showed a high expression of MUC5AC. Compared to the AM and SRAM groups, the CAM and CSRAM group showed a further significant increase in MUC5AC expression, confirming that cold stimulation could aggravate the severity of asthma in mice. Compared to the CSRAM group, the administration of DEX alone did not significantly reduce the high expression of MUC5AC. The combination of MCG and DEX could inhibit the expression of MUC5AC, and there was a dose-dependent effect in the co-administration groups.

The aforementioned histopathologic examination findings demonstrate that MCG combined with the DEX treatment was able to alleviate inflammation, cell infiltration, and goblet cell proliferation aggravated by cold stimulation in mice.

### 2.6. Combination of DEX and MCG Inhibited the Expression of MUC5AC and Enhanced the Expression of TJs

The RT-PCR analysis (Figure 5A) of the expression levels of MUC5AC in lung tissues showed that the expression level of MUC5AC in the CSRAM group was significantly increased compared to the CSC group. Compared to the CSRAM group, the CSRAM + DEX + MCG_H_ group was able to significantly down-regulate the high level of MUC5AC, which is consistent with the results of immunohistochemistry.

The RT-PCR (Figure 5A) and Western blot (Figure 5B,C) techniques were used to determinate the expression of TJs (ZO-1, Occludin, and Claudin-1) in lung tissues. The RT-PCR results showed that, compared to the CSC group, the expression of TJs in the CSRAM group was significantly decreased. Compared to the CSRAM group, the CSRAM + DEX + MCG_H_ group was able to significantly increase the expression of TJs. Western blotting showed a significant decrease in TJs expression in the CSRAM group compared to the CSC group. MCG intervention significantly increased TJs expression, which was consistent with the results of the RT-PCR. The above results indicate that the combination treatment of DEX and MCG could reduce the high expression of MUC5AC, alleviate mucus hypersecretion, and increase the expression of TJs, thereby repairing damage to the airway epithelial barrier.

### 2.7. Combination of DEX and MCG Regulated the Gut Microbiota and Increased the Synthesis of SCFAs

The richness and diversity of gut microbial species in the CSC, CSRAM, and CSRAM + DEX + MCG_H_ groups were evaluated using Alpha diversity (Figure 6A). According to the ACE and Chao1 indices, the species richness of gut microbiota in the CSRAM group was significantly reduced compared to the CSC group, while the number of species in the CSRAM + DEX + MCG_H_ group was significantly elevated compared to the CSRAM group. On the basis of the Shannon index, although there was no significant difference compared to the CSC group, species diversity was down-regulated in the CSRAM group, while species diversity was significantly dialed back in the CSRAM + DEX + MCG_H_ group compared to the CSRAM group. In the light of the Simpson index, there were no significant differences among these three groups. Overall, the above results suggest that the combination treatment of DEX and MCG could effectively increase the gut microbial richness and diversity in CSRA model mice.

The degree of similarity of mouse gut microbial communities was assessed by beta diversity. The results of principal coordinate analysis (PCoA) indicate that these three groups each had unique microbial community structures and distributions (Figure 6B). Compared to the CSRAM group, the microbial community of the CSRAM + DEX + MCG_H_ group was significantly closer to that of the CSC group, further suggesting that the combination treatment of DEX and MCG could regulate the composition and structure of the intestinal microbial community towards the CSC group.

The abundance of the top 10 species across the three groups was compared at the phylum and genus levels (Figure 6C). The results of species abundance at the phylum level show that the abundance of *Firmicutes* and *Patescibacteria* increased and the abundance of *Bacteroidota*, *Desulfobacterota*, *Deferribacterota*, and *Verrucomicrobiota* decreased in the CSRAM group compared to the CSC group, resulting in the disruption of the intestinal flora. Compared to the CSRAM group, the combination treatment of DEX and MCG shifted the levels of *Firmicutes*, *Bateroidota*, *Patescibacteria*, *Desulfobacterota*, *Deferribacterota*, and *Verrucomicrobiota* toward normal retraction. The results of species abundance at the genus level show that the abundance of *Lactobacillus*, *Ligilactobacillus*, and *Limosilactobacillus* were increased, and the abundance of *Odoribacter* and *Lachnospiraceae_NK4A136_group* were decreased in the CSRAM group compared to the CSC group. Compared to the CSRAM group, the combination treatment of DEX and MCG could regress them towards normal levels, including significantly decreasing the abundance of *Lactobacillus*, *Ligilactobacillus*, and *Limosilactobacillus* and increasing the abundance of *Odoribacter* and *Lachnospiraceae_NK4A136_group*.

The evolutionary branching maps of differentiated species in these three groups were established based on the LEfSe analysis (Figure 6D). Only one biomarker (*s_Adlercreutzia_muris*) in the CSRAM group was identified, suggesting that the abundance of a large number of gut microorganisms was reduced in the CSRAM group. The number of biomarkers in the CSRAM + DEX + MCG_H_ group reached 43, which suggests that the MCG intervention significantly enhanced the abundance of gut flora. To more accurately characterize the relationship between intestinal flora and the combination of DEX and MCG for alleviating CSRA, 28 differential flora at the family, genus, and species levels were used for the subsequent integrated analysis.

The probiotics *Odoribacter* and *Lachnospiraceae_NK4A136_group*, modulating the intestinal flora and enhancing immunity, are major producers of SCFAs [24]. Since they were significantly enriched in the CSRAM + DEX + MCG_H_ group, the contents of SCFAs in the fresh feces of mice from three groups were quantified (Figure 6E). The results demonstrated that the contents of acetic acid and propionic acid in the CSRAM group were significantly reduced compared to the CSC group (*p* < 0.01), and the content of butyric acid was also reduced, although there was no statistically significant difference. Compared to the CSRAM group, MCG in combination with DEX administration made a substantial improvement in the contents of acetic acid, propionic acid, and butyric acid.

### 2.8. Combination of DEX and MCG Regulated Gene Transcription Levels

The gene profiles of the lung tissues of the CSC, CSRAM, and CSRAM + DEX + MCG_H_ groups were analyzed by the PCA score plot (Figure 7A). Overall, in comparison to the CSRAM group, the CSRAM + DEX + MCG_H_ group was closer to the CSC group, suggesting that medication intervention resulted in a partial restoration of the gene profiles to normal levels. The DEG volcano maps (Figure 7B) were used to investigate the differential genes between the CSC group and the CSRAM group or between the CSRAM group and the CSRAM + DEX + MCG_H_ group. Pink dots represented the up-regulated genes (fold-change > 1.5, *p* < 0.01), whereas blue dots represented the down-regulated genes (fold-change < 0.67, *p* < 0.01). As shown in the VN plot (Figure 7C), compared to the CSC group, there were 2020 genes up-regulated and 1853 genes down-regulated in the CSRAM group; compared to the CSRAM group, there were 985 genes up-regulated and 705 genes down-regulated in the CSRAM + DEX + MCG_H_ group, among which there were 1564 genes tended to the levels of the CSC group (including 879 up-regulated genes and 685 down-regulated genes). The obtained differentially expressed genes (DEGs) were then classified and annotated using the GO and KEGG databases. The GO processes (biological processes, cellular components, and molecular functions) contained 30 annotations of functions with the highest enrichment score and were presented in a GO circle diagram (Figure 7D). The results show that the combination of DEX and MCG can regulate the immune system process, receptor complex, immune receptor activity, etc. The KEGG results show that DEGs were significantly enriched in the Rap1 signaling pathway, cGMP-PKG signaling pathway, PI3K-Akt signaling pathway, MAPK signaling pathway, Th17 cell differentiation, etc., and the pathways (*p* < 0.05 and impact > 0.1) potentially associated with glucocorticoid resistance and associated with the effects of DEX + MCG were presented in Figure 7E.

### 2.9. Combination of DEX and MCG Modulated Metabolite Levels

The lung metabolic profiles of the CSC, CSRAM, and CSRAM + DEX + MCG_H_ groups were analyzed by the PCA score plot (Figure 8A). Compared to the CSC group, the lung metabolic profile of the CSRAM group was aberrant. The CSRAM + DEX + MCG_H_ group was located between the CSC group and the CSRAM group, suggesting that the metabolic profile of the medication intervention tended towards normal levels. There was a maximal partition in the OPLS-DA score plot between the CSC group and the CSRAM group, or between the CSRAM group and the CSRAM + DEX + MCG_H_ group (Figure 8B). The permutation test confirmed the model’s good explanatory and predictive rates. The differential metabolites were investigated using differentially expressed metabolites (DEMs) volcano maps, with pink dots representing the up-regulated metabolites (fold-change > 1, *p* < 0.05, VIP > 1) and blue dots representing the down-regulated metabolites (fold-change < 1, *p* < 0.05, VIP > 1) (Figure 8C). As shown in the VN plot (Figure 8D), compared to the CSC group, the CSRAM group up-regulated 649 metabolite levels and down-regulated 546 metabolite levels; compared to the CSRAM group, the CSRAM + DEX + MCG_H_ group showed a re-regulation (tending to the levels in the CSC group) of 331 metabolites, including an up-regulation of 138 metabolite levels and a down-regulation of 193 metabolite levels. The enrichment analysis of DEMs was then performed, and the KEGG database was used to visualize the metabolic pathways. The results (Figure 8E) show that DEMs were enriched in 28 metabolic pathways, including purine metabolism, sulfur metabolism, pentose phosphate pathway, arachidonic acid metabolism, phenylalanine, tyrosine, and tryptophan biosynthesis, etc.

### 2.10. Multi-Omics Integrated Analysis

A total of 483 DEGs and 106 DEMs were screened in the correlation analysis between transcriptomics and metabolomics; 325 DEGs were screened in the correlation analysis between transcriptomics and microbiomics; and 84 DEMs were screened in the correlation analysis between transcriptomics and microbiomics. Among these, 110 DEGs were simultaneously associated with DEMs and flora, and 30 DEMs were simultaneously associated with DEGs and flora. A joint pathway analysis was conducted on 110 DEGs and 30 DEMs, and 15 metabolic/signaling pathways (including 37 DEGs and 7 DEMs) were screened as closely related to SRA. The relationships were visualized through the “gene–metabolite–pathway” network (Figure 9A). We then identified the eight flora that had an absolute correlation coefficient greater than 0.75 with both the 37 DEGs and the 7 DEMs. The correlations of eight flora with the DEGs and DEMs were visualized by heatmaps (Figure 9B,C). The analysis yielded compelling insights, demonstrating a profound association between the majority of the flora and DEGs or DEMs. Notably, the correlation between each flora and at least one DEG or DEM exceeded an absolute value of 0.75, indicative of robust connections. Next, we confirmed the significant contribution of seven flora, 34 DEGs, and 6 DEMs in the alleviation of CSRA with DEX in combination with MCG by ROC curve using an AUC > 0.80 and *p* < 0.05 as the criteria (Figure 9D). Finally, the Sankey diagram was used to show the “flora–gene–metabolite–pathway” network (Figure 9E). The results show that gut microorganisms, such as *g_GCA_900066575*, *g_Oscillibacter*, *g_Peptococcus*, and *g_Tuzzerella*; DEGs, such as *Adcy8*, *Adra1d*, *Pik3cd*, and *Plcb2*; and metabolites, such as 2′-deoxyguanosine-5′-monophosphate, adenosine, dihydroceramide, inosine, and ribose-1-phosphate, may be more closely associated with the combined treatment of CSRA with DEX and MCG. These biomarkers were significantly enriched in signaling pathways with potential modulatory effects on airway smooth muscle contraction, airway remodeling, and inflammatory responses, such as the cAMP signaling pathway, the cGMP-PKG signaling pathway, the calcium signaling pathway, and the PPAR signaling pathway. It was worth noting that these biomarkers were also enriched in the PI3K-Akt signaling pathway and the MAPK signaling pathway, closely associated with the development of steroid resistance, and coincided with our aim of exploring drug combination for the alleviation of CSRA. Consequently, we selected the PI3K-Akt and MAPK signaling pathways as focal points for further validation.

### 2.11. Combination of DEX and MCG Inhibits Activation of the PI3K-Akt/MAPK Pathway

Western blotting was used to determine the levels of PI3K, p-PI3K, Akt, p-Akt, p-38, and p-p38 proteins in lung tissues. The results (Figure 9F,G) show that there was no significant change in the levels of three total proteins (PI3K, Akt, and p38) in the CSC group, CSRAM group, and CSRAM + DEX + MCG_H_ group. Compared to the CSC group, the levels of p-PI3K, p-Akt, and p-p38 in the CSRAM group were significantly increased, suggesting that the PI3K-Akt/MAPK signaling pathway was activated by significantly increasing the phosphorylation levels of PI3K, Akt, and p38. Compared to the CSRAM group, the levels of p-PI3K, p-Akt, and p-p38 in the CSRAM + DEX + MCG_H_ group were significantly down-regulated, indicating that DEX combined with MCG could inhibit the PI3K-Akt/MAPK signaling pathway.

## 3. Discussion

Low temperatures greatly increase the risk of respiratory diseases, such as asthma and chronic obstructive pulmonary disease [25]. In our innovative research, we first investigated and validated the effects of low temperature on SRA. In our study, the decrease in Cydn and the increase in RI demonstrated that cold stimulation exacerbated airway hyperresponsiveness in asthma, or SRA, which is consistent with previous reports [12,13]. In addition, we also found that SRA has a greater response to cold stimulation compared to asthma, which may be related to its weaker regulatory effects on lung function and lower lung compliance.

Combined medication is an effective treatment for SRA. For example, clinical cohort studies have found that the combination of azithromycin and glucocorticoids could significantly reduce the frequency of steroid-resistant asthma attacks in patients [26]; in a mouse model, the combination of andrographolide and DEX enhanced antioxidant defense, reduced airway hyperresponsiveness, and restored sensitivity to steroids [27]; and the combination treatment of Rhodiola rosea and DEX could effectively alleviate the SRA symptoms by modulating immunity, microbiota, and metabolic disorders [28]. Based on our study, it could be concluded that combined medication is also an effective treatment for CSRA.

When inflammation occurs, neutrophils are attracted by chemotactic substances to the site of inflammation, playing a very important role in the non-specific cellular immune system [29]. Cold stimulation could lead to an increase in neutrophil count in human or dog peripheral blood or BALF and exacerbate neutrophil infiltration in horse lung tissues [30,31,32]. In SRA patients, a large accumulation of neutrophils in the airways and lungs is activated, leading to a sustained inflammatory response and airflow limitation [33]. Our findings in the SRA mouse model are consistent with those reported in the literature. Furthermore, we found that cold stimulation could increase the number and proportion of neutrophils in the BALF of SRA mice, which was again demonstrated by subsequent H&E staining results.

The Th1/Th17 immune imbalance may play an important role in the severity and steroid resistance of SRA [34]. IL-17 secreted by Th17 is the most important effector in the mobilization, recruitment, and activation of neutrophils, while IFN-γ secreted by Th1 inhibits neutrophil apoptosis [35,36]. This explains the fact that, in our work, compared to the normal group, the IFN-γ levels in the AM and CAM groups decreased, while the IFN-γ levels in the SRAM and CSRAM groups increased. The interaction between IL-17 and IFN-γ promotes the production of the chemokine CXCL8 and the cytokine IL-1β, thereby promoting neutrophil recruitment at the inflammation site and amplifying the local inflammatory response [37,38]. In addition, we found that cold stimulation was able to increase the levels of IL-1β, IFN-γ, and IL-17, which is consistent with previous reports [12,39,40]. We also confirmed that DEX combined with MCG was able to effectively reduce the levels of IL-17, IFN-γ, IL-1β, and CXCL8, suggesting that the co-administration could regulate the immune imbalance of Th1/Th17.

Mucus blockage and airway obstruction caused by an excessive secretion of mucus are the main causes of asthma [41]. Goblet cells are mucus-secreting epithelial cells in the respiratory tract, and continuous goblet cell hyperplasia is also an important feature of asthma. MUC5AC, the major mucin secreted by goblet cells, is associated with AHR and mucus plugging in a variety of chronic respiratory diseases [42,43]. The secretion of mucus, goblet cell proliferation, and expression of MUC5AC in mouse lungs have been investigated by AB-PAS staining and immunohistochemical staining. In our established mouse asthma model, the cold stimulation exacerbated mucus hypersecretion with goblet cell hyperplasia and elevated MUC5AC expression, which is consistent with the effects of cold stimulation on human bronchial epithelial cells, or AECOPD models [7,44]. MCG combined with DEX showed the ability to regulate the above-mentioned tendencies to a normal level.

TJs, mainly composed of transmembrane proteins Occludin and Claudin and the cytoplasmic scaffolding protein ZO-1, have selective permeability to macromolecular substances and ions [45]. They are components of the physical barrier of the bronchial epithelium. The biopsy results of bronchial epithelial cells in asthma patients showed that the transcriptional loss of ZO-1 and Occludin caused the increased permeability of the airway epithelial barrier [46]. Claudin-1 is confirmed to be closely associated with the disruption of epithelial structure in respiratory diseases and plays a key role in the pathological process of asthma [47]. In addition, neutrophils not only promote inflammation but also participate in regulating the function of the airway epithelial barrier and inhibiting the expression of TJs [48]. In our current work, we confirmed that the combination of MCG and DEX could significantly increase the expression of TJs and exert a significant protective effect on the airway epithelial barrier, which might relate to promoting the redistribution of TJs or reducing the infiltration and aggregation of neutrophils in lung tissues [49].

*Firmicutes* and *Bacteroidetes* are the microbial communities with the largest percentage of intestinal species, and their abundance ratio is regarded as a marker of intestinal microbial dysbiosis [50]. Propionic acid and butyric acid are the main short-chain fatty acids that are capable of restoring HDM-induced bronchial epithelial barrier dysfunction and are believed to be used for the treatment of asthma [51]. At the species level, we found that the abundance of *Roseburia* and *Butyricoccus* was high in the CSRAM group and low in the *Lachnospiraceae_NK4A136_group*. This is consistent with the effects on gut microbes that have been reported for OVA-based models of allergic airway inflammation [52], whereas the intervention of cold stimulation and NTHi led to more changes in the abundance of gut microbes. Our research results indicate that co-administration improved the imbalance between *Firmicutes* and *Bacteroidetes*, promoted the balance of the intestinal flora, significantly increased the content of SCFAs, and promoted the repair of lung epithelial dysfunction. This demonstrates that the combination therapy had a powerful regulatory effect on the intestinal flora and its metabolites.

PI3K is an intracellular lipid kinase, and Akt is an important downstream target. The phosphorylation levels of both proteins can be considered indicators of the activity of this signaling pathway [53]. Due to the deacetylation of the glucocorticoid receptor (GR) by HDAC2, its expression and activity decreases are associated with the development of SRA [54]. It has been shown that cigarette smoke inhalation or Haemophilus influenza infection could down-regulate HDAC2 activity by activating the PI3K-Akt signaling pathway, leading to steroid resistance in asthmatic mice [55,56].

MAPK belongs to the serine/threonine protein kinases family, and the p38MAPK pathway is one of the three MAPK signaling pathways highly correlated with the severity of asthma [57]. Our previous study found that the p38MAPK pathway in NTHi-infected asthma model mice exhibited overactivation, exacerbating the development of SRA [6]. At present, it has been found that the combination therapy significantly inhibits the levels of p-PI3K, p-Akt, and p-p38. Therefore, it is inferred that MCG combined with DEX improved steroid resistance by inhibiting the phosphorylation of the PI3K/Akt signaling pathway and the p38MAPK signaling pathway.

Although this study explored the effects of cold stimulation on SRA and the new application of MCG in the amelioration of CSRA, there are still some issues that need further in-depth research in the future. For instance, despite having quantitatively measured the content of 15 ginsenosides and total ginsenosides in MCG, we are still unable to pinpoint which specific ginsenosides play a critical role in this process. Given the structural similarity between numerous ginsenosides and steroid hormones, we hypothesize that the more common ginsenosides, as opposed to rare ginsenosides present in lower concentrations, may exert a more prominent effect. However, this necessitates further experimental validation in the future. While the total ginsenosides content, accounting for 81.45% of MCG, can be considered pivotal in improving CSRA, the remaining 18.55% of components still require identification to clarify whether they contribute to MCG’s ability to alleviate CSRA. Further exploration is needed to confirm the impact of MCG on downstream proteins of the PI3K/Akt or p38MAPK signaling pathways, such as HDAC2 or GR, thereby clarifying the role of MCG in modulating proteins closely associated with steroid resistance. Additionally, the beneficial effect of MCG on CSRA through the modulation of the gut microbiota necessitates validation via fecal microbiota transplantation experiments.

## 4. Materials and Methods

### 4.1. Materials and Reagents

Ovalbumin (OVA, A5503) was purchased from Sigma-Aldrich Trading Co., Ltd., Shanghai, China. Alum adjuvant (A10853) was obtained from Beijing Jinming Biotechnology Co., Ltd., Beijing, China. Dexamethasone sodium phosphate injection (220420) was produced by Shanghai Full Woo Biotechnology Co., Ltd., Shanghai, China. Non-typeable Haemophilus influenza (NTHi) was derived from clinical samples [6]. ELISA kits of IL-1β, IL-10, IL-17, CXCL8, IFN-γ, and LPS were all purchased from FeiYa Biotechnology Co., Ltd., Yancheng, China. Both PerCP Anti-Mouse CD11b (E-AB-F1081F) and APC Anti-Mouse Ly6G (E-AB-F1108E) antibodies were purchased from Elabscience Biotechnology Co., Ltd., Wuhan, China. MUC5AC (MA5-12178) antibody was provided by Thermo Fisher Scientific, Waltham, MA, USA. ZO-1 (21773-1-AP), Occludin (66738-1-Ig), and Claudin-1 (28674-1-AP) antibodies were all provided by Proteintech Group, Inc. (Rosemont, IL, USA). PI3K (BS-10657R) and Phospho-PI3K (BS6417R) antibodies were provided by Bioss Antibodies, Beijing, China. p38 (BF8015), Phospho-p38 (AF4001), Akt (AF6261), and Phospho-Akt (AF0016) antibodies were all provided by Affinity Biosciences, Changzhou, China. HPLC-grade methanol and acetonitrile were provided by Fisher Chemical Company, Waltham, MA, USA.

### 4.2. Preparation and Analysis of MCG

The 15-year-old mountain-cultivated ginsengs were collected from a cultivation area in Fusong, China. The ginseng was identified by our group according to the 2020 Edition of the Chinese Pharmacopoeia (ChP). The voucher specimen (No. 220928) was preserved in the Natural Drug Research Center at Jilin University, Changchun, China. The dried mountain-cultivated ginseng was cut into thick slices and then extracted three times (each time for 24 h) with 10 times the volume of 80% ethanol. The extracts were combined, and the ethanol was recovered to obtain a concentrated solution, which was freeze-dried to obtain the 80% ethanol extract of mountain-cultivated ginseng (MCG) powder. The total ginsenosides in MCG were quantitatively determined according to the methods specified in the 2020 Edition of the Chinese Pharmacopoeia (ChP). The quantitative chemical analysis of 15 monomeric ginsenosides in MCG was performed using a Waters Acquity ultra-performance LC (including Binary Solvent Manager, Sample Manager, and TUV Detector) and the Empower 3 software (database version: 7.10) (Waters, Manchester, UK). The chromatographic column was an Acquity UPLC BEH C18 column (2.1 mm × 100 mm, 1.7 μm) with the mobile phases of water (A) and acetonitrile (B). Gradient elution: 0.0~5.0 min, 19% B; 5.0~8.0 min, 19~21% B; 8.0~11.0 min, 21~28% B; 11.0~17.0 min, 28~31% B; 17.0~19.0 min, 31~35% B; 19.0~22.0 min, 35~45% B; 22.0~30.0 min, 45~100% B; detection wavelength: 203 nm; flow rate: 0.4 mL/min; column temperature: 30 °C; and injection volume: 5 μL. For the solution preparation, MCG was dissolved in methanol to obtain a test solution with a concentration of 6.0 mg/mL. The reference substances of ginsenoside Rg1, Re, PPD, Rg2, Rb1, Rc, Rb2, Rd, Rg6, F4, Rh4, (20S)-Rg3, PPT, Rk1, and Rf were accurately weighted and dissolved in methanol to prepare the mixed stock reference solution, which was then diluted to series concentrations to establish the calibration curves.

### 4.3. Animals and Grouping

Specific pathogen-free (SPF) female BALB/c mice (20 ± 2 g) were obtained from YISI Experimental Animal Technology Co., Ltd., Changchun, China. The animals were housed in the Laboratory Animal Center of the College of Basic Medical Sciences of Jilin University, with free access to food and water and a day/night interval of 12 h. The normal ambient temperature was 25 ± 2 °C with humidity of 60 ± 5%. The cold-stimulated ambient temperature was 10 ± 2 °C with humidity of 35 ± 5%. All the procedures were approved by the experimental animal ethics committee of the School of Basic Medical Sciences of Jilin University (NO. 2024329).

We optimized the conditions to construct mouse models based on previous reports in the literature [6,12]. The mice were provided a week of acclimation at normal ambient temperature before being randomly assigned to 10 groups (n = 15) as follows: normal control (NC) group, cold-stimulated control (CSC) group, asthma model (AM) group, cold-stimulated asthma model (CAM) group, steroid-resistant asthma model (SRAM) group, cold-stimulated steroid-resistant asthma model (CSRAM) group, CSRAM + DEX (2 mg/kg/d) group, CSRAM + DEX (2 mg/kg/d) + MCG_L_ (50 mg/kg/d) group, CSRAM + DEX (2 mg/kg/d) + MCG_M_ (100 mg/kg/d) group, and CSRAM + DEX (2 mg/kg/d) + MCG_H_ (200 mg/kg/d) group. On days 0, 7, and 14, the mice in the NC group and the CSC group were intraperitoneally injected with 10 mL/kg of sterile saline. All other groups of mice were intraperitoneally injected with 10 mL/kg of a 0.5 mg/mL OVA-sensitizing solution. From day 22 to day 26, the mice in the NC group and the CSC group were nebulized with sterile saline, and all other groups of mice were nebulized with 5% OVA for 30 min. Following the last nebulization on day 26, 10 mL/kg of PBS was injected into each mouse’s trachea in the NC group and CSC group, and an equivalent volume of NTHi was injected into the other group’s tracheas. From day 27 to day 31, DEX injection and MCG were administered by intraperitoneal injection (10 mL/kg) and intragastric administration (10 mL/kg), respectively. Each control and model group was administered with equal doses of sterile saline. The cold stimulation was started from day 21 to day 31, and the frequency was twice a day for 3 h in the morning and 3 h in the afternoon. On day 32, all mice were sacrificed after fasting for 12 h, and all the specimens were collected. The experimental procedures were shown in Figure 10.

### 4.4. Pulmonary Function Examination

The mice (n = 3) were administered an intraperitoneal injection of 1% sodium pentobarbital to induce anesthesia. The nebulized acetyl methacholine solution was then introduced at 5 concentrations (0, 3.125, 6.25, 12.5, and 25 mg/mL, each for 3 min) after inserting the cannula into the mouse’s airway. The airway hyperresponsiveness, including resistance of inspiratory (RI) and dynamic respiratory compliance (Cdyn), after each nebulization dosage of acetyl methacholine, was then measured using the DSI Buxco RC System (Fine Point, Wayland, MA, USA) according to a previously published technique [58]. The results are presented as the mean maximal resistance minus the baseline resistance.

### 4.5. Neutrophil Counting in the Bronchoalveolar Lavage Fluid (BALF)

The eyeballs of all the groups (n = 3) were removed for blood collection, and then, the lung tissues of all the groups (n = 3) were lavaged three times with pre-cooled saline. BALF was gathered and centrifuged (300× *g*, 4 °C, 5 min). The supernatant was used for the ELISA, and the precipitate fraction was used for the neutrophil counting. Following the erythrocyte cleavage, washing, APC-Ly6G and PE-CD11b antibody labeling, and single-cell suspension preparation, flow cytometry (NovoCyte, ACEA, Santa Clara, CA, USA) was used to assess the quantity and percentage of Ly6G^+^- and CD11b^+^-labeled neutrophils [6,59].

### 4.6. Detection of LPS and Cytokines in Serum and BALF

The serum (obtained from whole blood by centrifuging at 3000 rpm for 20 min at 4 °C) and the BALF supernatant were used to perform ELISA in order to investigate the effects of all groups (n = 3) on lipopolysaccharides (LPS, a component of bacterial cell walls), pro-inflammatory cytokines (IL-1β and IFN-γ), anti-inflammatory cytokines (IL-10), and cytokines (CXCL8 and IL-17, intimately linked to neutrophil aggregation). ELISA kits were used to determine the above-mentioned.

### 4.7. Histopathologic Examination of Lung Inflammation and Mucus Hypersecretion

The essential markers to evaluate the severity of asthma pathology are mucus hypersecretion, inflammatory cell infiltration, bronchial thickness, and goblet cell hyperplasia [60]. Additionally, Mucin 5AC (MUC5AC) is highly correlated with airway neutrophils, declining lung function, and asthma exacerbations [61]. The lung tissues of all groups (n = 3) were washed with saline and preserved in 4% paraformaldehyde for 24 h. For histopathological staining, lung tissues were then sectioned and fixed in paraffin. We used H&E staining to investigate the effects of MCG + DEX on inflammatory cell infiltration and bronchial thickening in the lung tissues of mice, as well as AB-PAS staining to examine the co-administration’s effects on goblet cell proliferation in the mice’s lung tissues [62]. For immunohistochemical analysis, to evaluate the effects on mucus hypersecretion and bronchial epithelial cell mucin, pre-dehydrated and waxed lung tissue sections incubated with anti-MUC5AC antibodies were used to determine MUC5AC [63]. The histopathological changes were observed under an optical microscope and photographed.

### 4.8. Detection of MUC5AC and Tight Junction Protein (TJ) Expression in Lung Tissues

Severe steroid-resistant asthma could induce a massive loss of TJs [64]. The lung tissues (n = 3) of the CSC group, CSRAM group, and CSRAM + DEX + MCG_H_ group were collected. Both RT-PCR and Western blot were used to determine the expression of target proteins (ZO-1, Occludin, and Claudin-1) in lung tissues at the mRNA and protein levels. Furthermore, we used RT-PCR to evaluate the expression level of MUC5AC in lung tissues. As for RT-PCR, total RNA was isolated from the lung tissues of each group, and cDNA was then reverse-transcribed according to the kit instructions. The amplification was performed using cDNA as a template on a fluorescent quantitative PCR device (Bio-Rad, Hercules, CA, USA). The sequences of primers are presented in Appendix A. As for the Western blot analysis, lung tissue blocks were prepared, and protease inhibitors were added (Beyotime, Shanghai, China). The protein contents were determined using a BCA kit (Beyotime, China). Total proteins were separated in 12% gels using SDS-PAGE (Biosharp, Hefei, China) and then put onto PVDF membranes (Millipore, Bedford, MA, USA). After chemiluminescence was finished, the optical density values of the target bands were examined using the AlphaEase FC Version 4 software.

### 4.9. Analysis of Microbiomics and Microbial Metabolites

To explore the differences in microbial diversity in the intestinal contents of mice, a single-molecule real-time sequencing (SMTS) method based on the PacBio sequencing platform was used to sequence the marker genes for third-generation microbial diversity; then, circular consensus sequencing (CCS) sequences were clustered, filtered, or denoised; finally, species annotation and abundance analysis were conducted [65]. The intestinal contents (n = 6) of the CSC group, CSRAM group, and CSRAM + DEX + MCG_H_ group were collected. The microbiomics test was performed by Biomarker Technologies Co., Ltd., Beijing, China. Briefly, full-length primer sequences used to create individual primers with barcodes were amplified by PCR and purified; the whole DNA of the mouse intestinal contents was isolated and identified; the PCR products were measured and homogenized to create a sequencing library (SMRT Bell) following quality verification; sequencing was then carried out using PacBio Sequel II. UCHIME v4.2, cutadapt 1.9.1, and Lima v1.7.0 was used to analyze the acquired data.

Short-chain fatty acids (SCFAs), the microbial metabolites produced by intestinal flora, could reduce bacterial translocation, preserve intestinal integrity, and maintain intestinal homeostasis. They are an important source of energy for the host and microorganisms [66]. The identification and quantification of SCFAs in fresh mouse feces were performed using Thermo Trace 1300 (Thermo Fisher Scientific, Waltham, MA, USA), and 4-methylpentanoic acid was used as an internal standard [67,68]. The chromatographic and mass spectrometric settings are detailed in Appendix A.

### 4.10. Analysis of Transcriptomics

In order to explore gene expression and gene structure on a broad scale and to discuss the molecular mechanism of DEX + MCG, transcriptome sequencing technology was used to obtain transcribed mRNA sequence data based on previous studies [69]. After the mice were euthanized, lung tissues from the CSC group, CSRAM group, and CSRAM + DEX + MCG_H_ group (n = 6) were immediately taken and rapidly frozen in liquid nitrogen for 3 h before being transferred to −80 °C for storage. The RNA sequence detection was also performed by Biomarker Technologies Co., Ltd. The main steps are as follows: Firstly, total RNA was extracted from lung tissues; mRNA was purified from total RNA using poly-T oligo-attached magnetic beads (Beckman Coulter, Brea, CA, USA); the fragmented mRNA served as a template for the synthesis of the first and second strands of cDNA; and the double-stranded cDNA was purified, end-repaired, A-tailed, and ligated to sequencing junctions. Secondly, fragment size selection was performed using AMPure XP beads, and the cDNA library was enriched using PCR. The libraries were processed in PE150 mode on the Illumina NovaSeq6000 sequencing platform once quality control was completed. Thirdly, clean data, obtained by filtering sequencing data, were aligned with the designated reference genome. Finally, bioinformatic analyses, including structure-level analysis, differential expression gene (DEG) analysis, gene function annotation, and function enrichment, were performed after the mapped data were acquired.

### 4.11. Analysis of Non-Target Metabolomics

To examine the overall dynamic changes in metabolites in the mice, non-target metabolomics was performed using pre-prepared mouse lung tissues. The metabolomics test was also performed by Biomarker Technologies Co., Ltd. A Waters Acquity I-Class PLUS UPLC system connected to a Waters Xevo G2-XS QTOF mass spectrometer (Waters Co., Milford, MA, USA) was used to perform detections via an electrospray ionization interface. Lung samples (n = 6) of the CSC, CSRAM, and CSRAM + DEX + MCG_H_ groups were collected. An extraction solution was prepared with methanol and acetonitrile in a volume ratio of 1:1. The 50 mg of lung sample and 1000 μL of extraction solution (containing 20 mg/L of internal standard) were vortexed for 30 s, ground, and sonicated. The supernatant after standing and centrifugation was taken for vacuum-drying [70]. Subsequently, the extract was reconstituted and tested on the machine [71]. The conditions for mass spectrometry and chromatography are described in detail in Appendix A. The analyses, which included a principal component analysis (PCA), orthogonal projections to latent structures discriminant analysis (OPLS-DA), and permutation tests of metabolites, were performed using MassLynx V4.2, Progenesis QI (Waters Co., Milford, MA, USA). The differentially expressed metabolites (DEMs) with a fold-change > 1, *p* < 0.05, and VIP > 1.0 were considered as potential biomarkers. The DEMs were imported into the MetaboAnalyst 6.0 database (http://www.metaboanalyst.ca (accessed on 27 February 2024)) to analyze the enriched metabolic pathways, and the figures presented were all plotted on BMKCloud (http://www.biocloud.net (accessed on 2 March 2024)), Chiplot (http://www.chiplot.online (accessed on 5 March 2024)), and bioinformatics (https://www.bioinformatics.com.cn (accessed on 5 March 2024)).

### 4.12. Integrated Multi-Omics Analysis

Step 1: Correlation analysis between transcriptomics and metabolomics: The DEGs and DEMs, closely related to the effectiveness of DEX + MCG, were imported into the MetaboAnalyst database, and a joint pathway analysis was performed. Step 2: Correlation analysis between any two omics: The FPKM of DEGs (or the quantitative values of DEMs) and the relative abundance of differential bacteria were calculated using the Spearman correlation algorithm based on the R language, respectively. DEGs (or DEMs) with coefficients > 0.75 or <−0.75 were considered to be closely related to the intestinal flora. DEGs and DEMs with coefficients > 0.90 or <−0.90 were considered to be closely related. Step 3: Taking the intersection of the selected DEGs (or DEMs) from steps 1 and 2, the obtained DEGs (or DEMs) were considered to be simultaneously associated with DEMs (or DEGs) and intestinal flora. Step 4: The DEGs and DEMs obtained from the screening of step 3 were subjected to joint pathway analysis; 15 pathways were screened for association with SRA, and the key DEGs and DEMs were identified. Step 5: The flora closely related to DEGs (or DEMs) in step 4 were taken to intersect with the yield flora, which were considered to be closely related to both DEGs and DEMs. Then, these flora with DEGs or DEMs in step 4 were correlated. Step 6: The flora closely related to DEGs and the flora closely related to DEMs in step 5 were taken to intersect with the yield flora considered to be closely related to both DEGs and DEMs. These flora with DEGs or DEMs were correlated to obtain biomarkers. Step 7: To verify whether the screened biomarkers could contribute to the treatment of CSRAM by DEX combined with MCG, a ROC curve analysis was performed for each biomarker with an AUC >0.80 and *p* < 0.05. The data obtained were used to construct the “flora–gene–metabolite– pathway” network using the Sankey diagram.

### 4.13. Verification of Key Pathways and Targets

Based on the results of our integrative analysis, the total protein and phosphorylation levels of the pathways and their downstream key targets that are closely related to MCG combined with DEX for the treatment of CSRA were selected for validation through Western blotting analysis.

### 4.14. Statistical Analysis

Non-omics data were statistically analyzed using the GraphPad Prism 9.5.1 software. All data are expressed as mean ± standard deviation (SD). Comparisons between two groups were conducted with an unpaired two-tailed Student’s *t*-test, and multiple comparisons were made between groups using a one-way analysis of variance (ANOVA) to assess the significance of the differences between parameters (*p* < 0.05). Multi-omics data were statistically analyzed using the BMKCloud (www.biocloud.net (accessed on 10 March 2024)) platform.

## 5. Conclusions

In the current study, a CSRA mouse model of allergic airway inflammation mimicking human asthma was successfully established for the first time, and the alleviating effects of 80% ethanol extract of mountain-cultivated ginseng (MCG) on CSRA was deeply investigated. The study found that MCG combined with DEX could significantly alleviate the CSRA-related symptoms, which could not be improved by DEX alone. Specifically, MCG was able to assist in alleviating the severe impairment of lung function, inhibiting the cascade of pulmonary inflammation, and mitigating bronchial epithelial physical barrier damage and mucosal barrier abnormalities. Mechanistically, the integrated omics analysis of microbiomics, transcriptomics, and metabolomics showed that seven gut microbes, 34 genes, six metabolites, and the involved 15 metabolic/signaling pathways (including the PI3K-Akt/MAPK signaling pathway closely related to steroid resistance) played a key role in the alleviation of CSRA by MCG. In short, MCG could effectively ameliorate CSRA by regulating the interactions among microbiota, genes, and metabolites. This study provides a new idea for the prevention and treatment of CSRA.

## Figures and Tables

**Figure 1 ijms-25-09110-f001:**
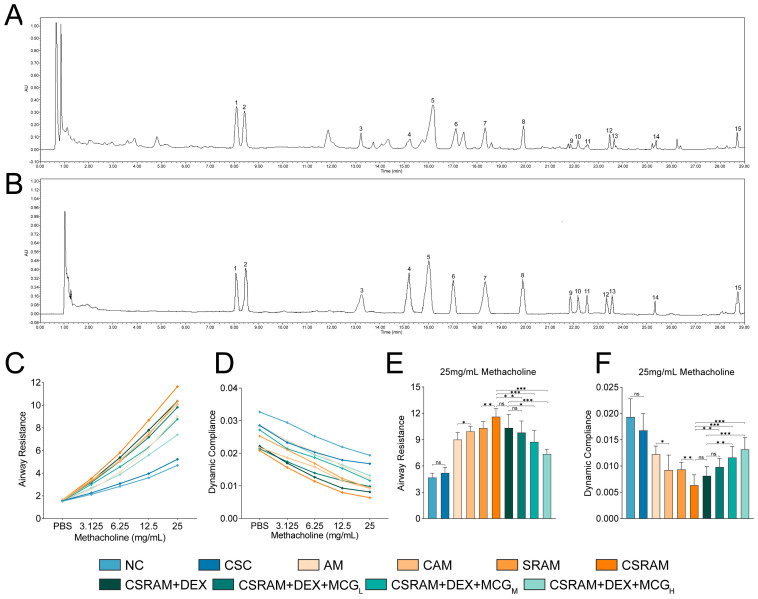
Chemical content analysis of MCG and pulmonary function examination. (**A**,**B**) Chromatograms of samples. (**A**) MCG test solutions. (**B**) Standard samples. 1. Rg1; 2. Re; 3. PPD; 4. Rg2; 5. Rb1; 6. Rc; 7. Rb2; 8. Rd; 9. Rg6; 10. F4; 11. Rh4; 12. (20S)-Rg3; 13. PPT; 14. Rk1; 15. Rf. (**C**) RI in response to increasing concentrations of methacholine. (**D**) Cdyn in response to increasing concentrations of methacholine. (**E**) RI in response to 25 mg/mL methacholine. (**F**) Cdyn in response to 25 mg/mL methacholine. Data are presented as mean ± SD. (ns, non-significant; * *p* < 0.05; ** *p* < 0.01; *** *p* < 0.001).

**Figure 2 ijms-25-09110-f002:**
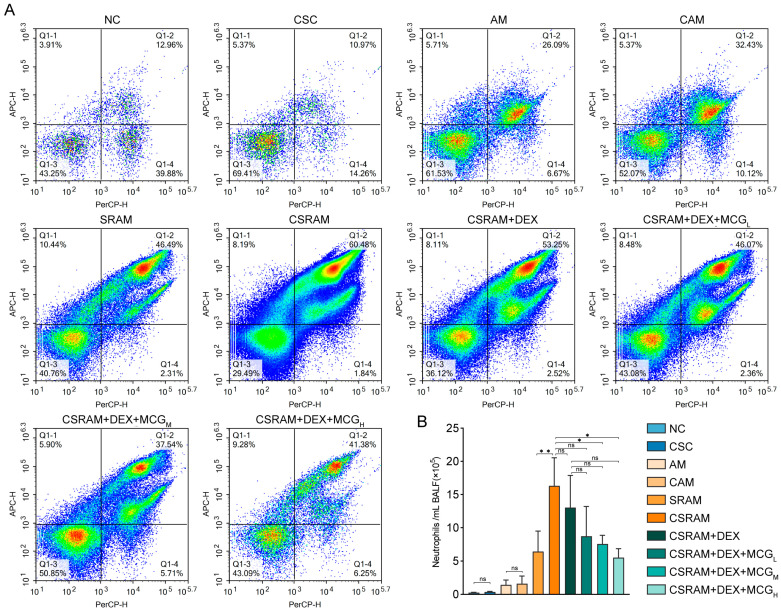
Quantifying neutrophils in mouse BALF with flow cytometry. (**A**) Proportion of neutrophils. (**B**) Count of neutrophils. Data are presented as mean ± SD. (ns, non-significant; * *p* < 0.05; ** *p* < 0.01).

**Figure 3 ijms-25-09110-f003:**
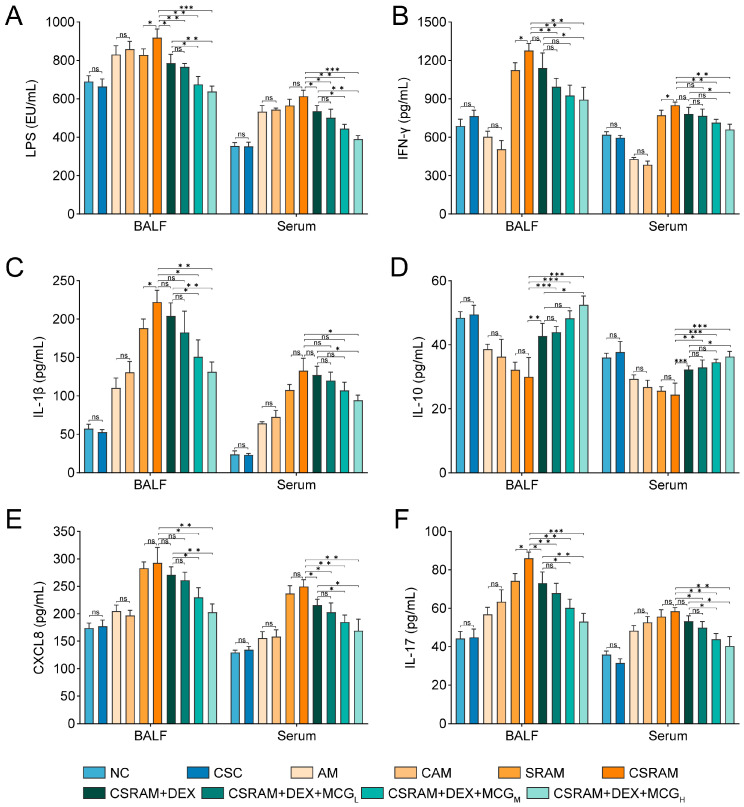
Effects of cold stimulation and co-administration on the levels of LPS and cytokines in the serum and BALF. (**A**) LPS. (**B**) IFN-γ. (**C**) IL-1β. (**D**) IL-10. (**E**) CXCL8. (**F**) IL-17. Data are presented as mean ± SD. (ns, non-significant; * *p* < 0.05; ** *p* < 0.01; *** *p* < 0.001).

**Figure 4 ijms-25-09110-f004:**
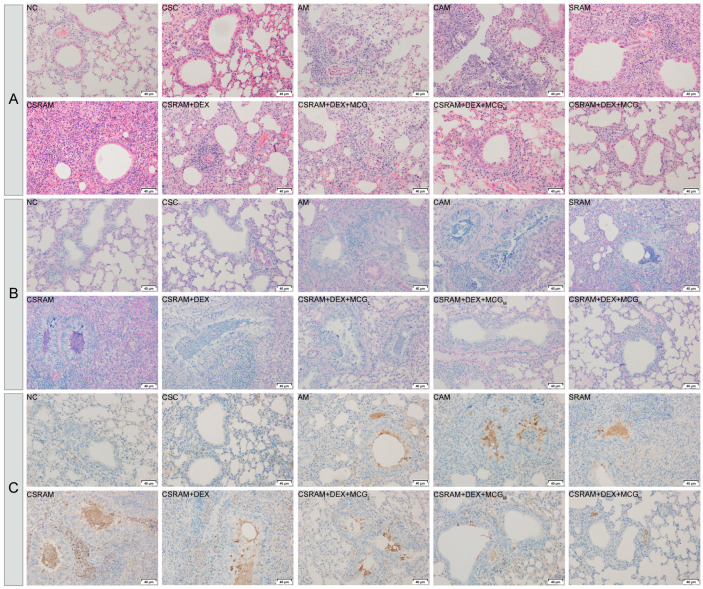
Effects of cold stimulation and co-administration on lung inflammation and mucus secretion. (**A**) H&E staining. (**B**) AB-PAS staining. (**C**) Immunohistochemical staining.

**Figure 5 ijms-25-09110-f005:**
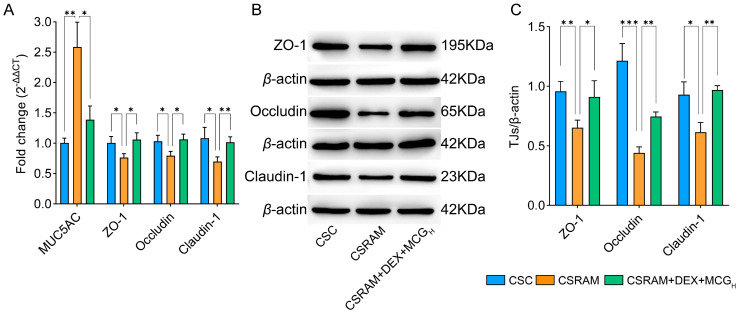
Effects of co-administration on MUC5AC and TJs in lung tissues. (**A**) RT-PCR assay to measure gene expression of MUC5AC and TJs. (**B**) Representative western blotting images of TJs. (**C**) Densitometric quantification of TJs expression. Reference: β-actin. (* *p* < 0.05; ** *p* < 0.01; *** *p* < 0.001).

**Figure 6 ijms-25-09110-f006:**
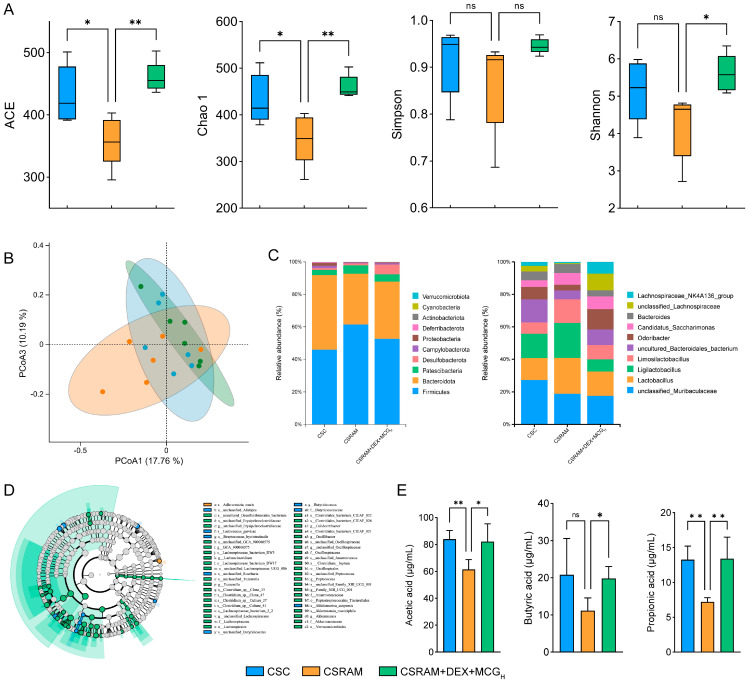
Combined treatment of DEX and MCG improved gut flora structure and increased the production of SCFAs. (**A**) Alpha diversity. (**B**) PCoA score. (**C**) Abundance structure of flora at the phylum and genus levels. (**D**) Evolutionary tree diagram of LEfSe. (**E**) Changes in the content of major SCFAs. Data are presented as mean ± SD. (ns, non-significant; * *p* < 0.05; ** *p* < 0.01).

**Figure 7 ijms-25-09110-f007:**
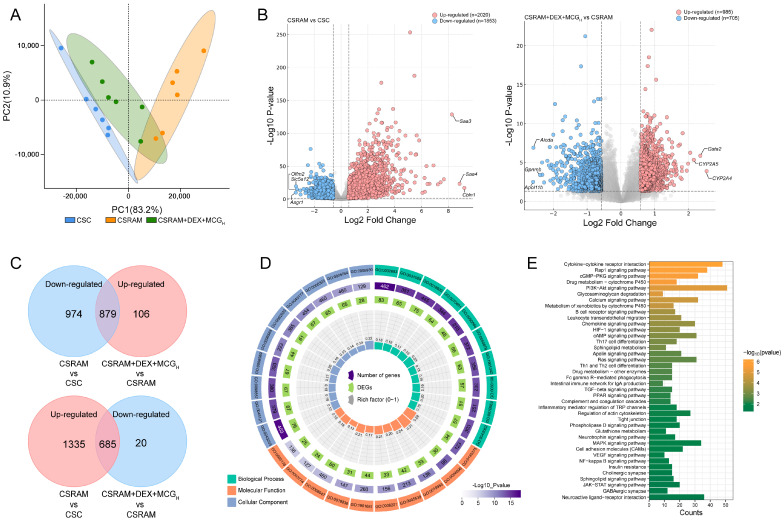
Effects of combined treatment of DEX and MCG on gene expression in lung tissues. (**A**) PCA score. (**B**) The distribution of DEGs. (**C**) VN maps of DEGs. (**D**) GO circle for 30 annotations of functions with the highest enrichment score. (**E**) KEGG pathway enrichment analysis.

**Figure 8 ijms-25-09110-f008:**
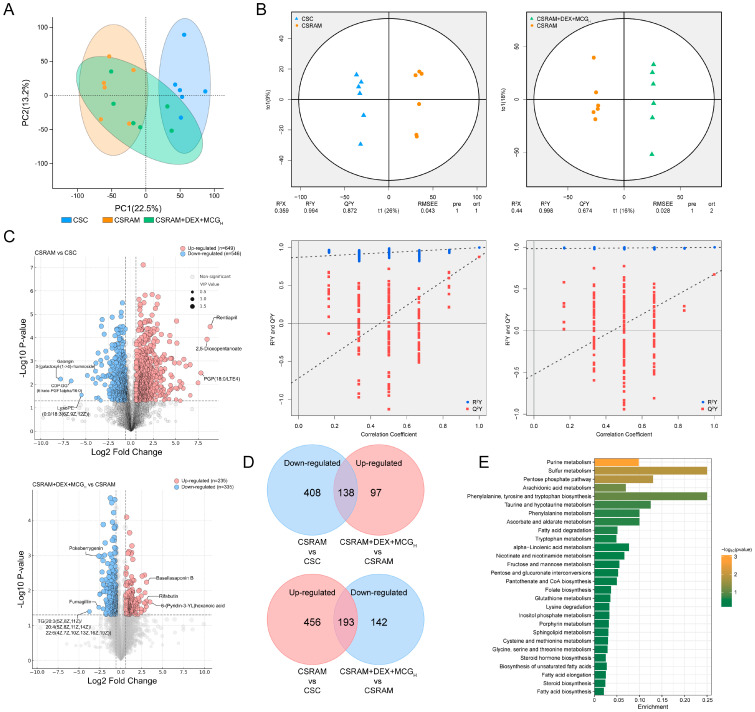
Effects of the combined treatment of DEX and MCG on the metabolite level in lung tissues. (**A**) PCA score. (**B**) OPLS-DA and permutation test. (**C**) The distribution of DEGs. (**D**) VN maps of DEMs. (**E**) KEGG pathway enrichment analysis.

**Figure 9 ijms-25-09110-f009:**
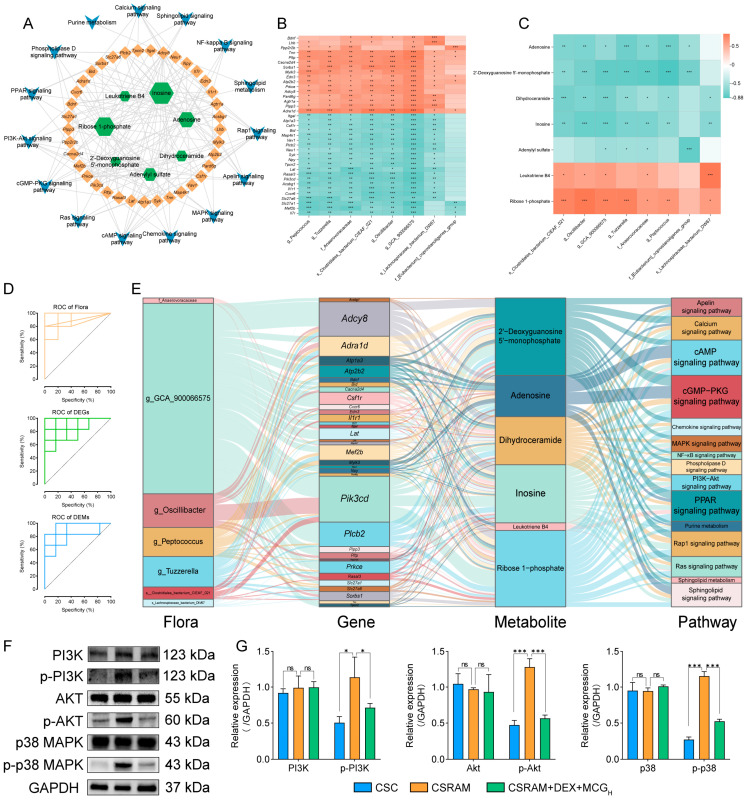
Integrated analysis of potential mechanisms and validation of PI3K-Akt/MAPK signaling pathway in the combination of DEX and MCG for the treatment of CSRA. (**A**) Correlation network of DEGs and DEMs. (**B**) Heatmap of correlation between DEGs and flora. (**C**) Heatmap of correlation between DEMs and flora. (**D**) ROC curve analysis. (**E**) Flora–gene–metabolite–pathway network. (**F**) Representative western blotting images. (**G**) Densitometric quantification of protein expression. Reference: GAPDH. (ns, non-significant; * *p* < 0.05; ** *p* < 0.01; *** *p* < 0.001).

**Figure 10 ijms-25-09110-f010:**
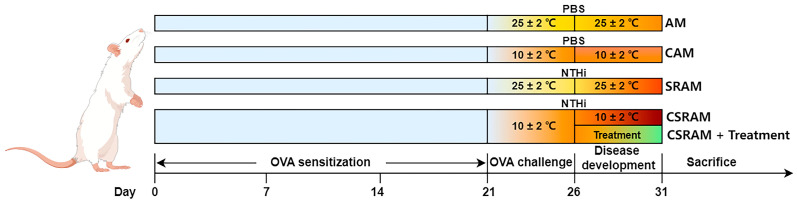
Procedures of the experiment.

## Data Availability

Multi-omics data from this study are deposited in the Science Data Bank databases: https://doi.org/10.57760/sciencedb.09798 (accessed on 30 June 2024) (microbiomics), https://doi.org/10.57760/sciencedb.09802 (accessed on 30 June 2024) (transcriptomics), and https://doi.org/10.57760/sciencedb.09800 (accessed on 30 June 2024) (metabolomics).

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
