# Peer review of "Integrated Multi-Omics Analysis Reveals Mountain-Cultivated Ginseng Ameliorates Cold-Stimulated Steroid-Resistant Asthma by Regulating Interactions among Microbiota, Genes, and Metabolites"

_ijms, 2024, doi:10.3390/ijms25169110_

Round 1

Reviewer 1 Report

Comments and Suggestions for Authors

The "flora-genes-metabolites-pathways" multidimensional network was established by Authors, using the multiomics integrated analysis to explore the potential mechanism in SRA treatment. This study provides relevant experimental data to elucidate the alleviating effects of MCG and DEX combined therapy on cold-stimulated SRA in mice, but with prospects for wider use of the results obtained. I highly appreciate the enormous amount of work, the interdisciplinarity of the whole experiment and the interesting presentation of results and conclusions, but I have some questions.

-        What were the conditions of UPLC analysis? There are also lack of information about type of chromatograph and column model, flow rate, composition of phases and qualitative determinants. Please, add this information to the manuscript.

-        Section 2 - there are very few literature references here. How was the methodology for the experiments described obtained?

-        Do Authors know which of  ginsenosides may be crucial in the experiment?

-        What assurance do we have that the ginsenosides have a key role in the CSRA ameliorates?

-        Can the results obtained be somehow related to human therapies?

Yours sincerely,
Reviewer

Author Response

我们衷心感谢各位提出的建设性意见。根据评论,对手稿进行了仔细修订。

 评论1 UPLC分析的条件是什么?此外,还缺乏关于色谱仪和色谱柱型号类型、流速、相组成和定性决定因素的信息。请将此信息添加到手稿中。
回应非常感谢您的提醒。我们已将UPLC条件添加到手稿的2.2部分。

评论 2第 2 部分 - 这里的文献参考很少。所描述的实验方法是如何获得的?    

回应在第 2 节中,我们描述的方法来自文献参考和我们对先前方法的改进。经过严格检查,我们已将必要的引用添加到修订版本中。

 评论3作者是否知道哪种人参皂苷在实验中可能至关重要?  

回应根据文献[1],如下图所示,人参皂甙在结构上与类固醇激素相似。我们认为,我们鉴定出的多种常见的人参皂苷可能共同发挥关键作用,因为它们的含量百分比较高(69.20%)。当然,不能排除含量较低的稀有人参皂苷起关键作用的可能性,因为它们也包含在我们准备的MCG中,但这还需要在未来进一步确定。

引用

  1. 李, J.;赵,J.;王旭;林,Z.;林,H.;Lin, Z. 人参皂苷 – 一种有前途的天然活性成分,具有甾体激素活性。食品与功能 2024, 15, 1825-1839.评论4我们有什么保证人参皂苷在CSRA改善方面发挥关键作用?

响应:非常感谢您的提醒。我们在讨论部分添加了与评论 3评论 4 相关的内容。

首先,先前的研究报道,各种人参皂甙在缓解哮喘[1-6]或改善类固醇耐药性方面具有药理作用[7-10]。其次,在CSRA的治疗中,单独使用地塞米松的效果不理想,而其与MCG合用能有效提高治疗效果。最后,定量测定了15种人参皂苷的含量,为69.20%,而MCG中人参皂苷的总含量达到81.45%。因此,可以合理地得出结论,人参皂苷作为 MCG 中的主要活性成分,在缓解 CSRA 方面起着关键作用。

引用

  1. 玉木,J.,J.中田,K.川谷,E.田谷和A.永井。2009. “人参皂苷通过释放一氧化氮诱导人支气管平滑肌松弛”,英国药理学杂志,130:1859-64。
  2. Lee、In-Seung、InJoon Uh、Ki-Suk Kim、Kang-Hoon Kim、Jiyoung Park、Yumi Kim、Ji-Hoon Jung、Hee-Jae Jung 和 Hyeung-Jin Jang。2016. '人参皂苷Rg3通过NF-κB通路在A549细胞和人哮喘肺组织中的抗炎作用', 免疫学研究杂志, 2016: 1-11.
  3. Chen, Tong, Lu Xiao, Lingpeng Zhu, Shiping 马, Yan 天华, 和 Hui Ji.2015. “人参皂苷 Rb1 在过敏性哮喘小鼠模型中通过降级 Th1/Th2 的抗哮喘作用”,炎症,38:1814-22。
  4. 李, 强, 翟纯苗, 王国栋, 贾周, 李伟光, 谢立群, 石占利.2021. “人参皂苷 Rh1 通过调节 Th1/Th2 细胞因子平衡减轻卵清蛋白诱导的哮喘”,生物科学、生物技术和生物化学,85:1809-17。
  5. 任, 苏梅, 刘瑞琪, 王玉杰, 丁宁, 和李颖霞. 2019.“人参皂苷化合物K类似物作为一类新型抗哮喘剂的合成和生物学评价”,《生物有机与药物化学快报》,29:51-55
  6. 徐、张、李亮昌、王重阳、江敬志、李丽、朱连华、金珊、金哲虎、李荣俊、李冠豪、严光海。2022. “G-Rh2 通过 AKT-Nrf2/NF-κB 和 MAPK-Nrf2/NF-κB 通路对肥大细胞介导的过敏反应的影响”,人参研究杂志,46:550-60。
  7. 牛,C.S.;叶,CH;叶,MF;Cheng, J. T. 人参皂苷 (Rh2) 通过激活糖皮质激素受体增加 3T3-L1 细胞中的脂肪生成。激素与代谢研究 2008, 41 (04), 271-276.
  8. 李, J. D., J ;刘,D;程, BB ;方,FF ;翁,L;王,C; 玲,CQ。人参皂苷 Rh1 通过逆转地塞米松诱导的耐药性,增强地塞米松对慢性炎症性疾病的抗炎作用。关节炎研究与治疗 2014, 16 (3).
  9. 冯,Y.;王,C.;程,S.;王旭;孟,X.;李,L.;杜,J.;刘清;郭英;孟,Y.;等人参皂苷 Rh1 改善地塞米松对 MRL/lpr 小鼠自身抗体产生和淋巴细胞增殖的影响。循证补充和替代医学 2015, 2015, 1-8.
  10. 高玲玲;朱,S.;李, J.;李, J.;张Z.;夏,C.;亨,Y.;张,M.;胡,J.;魏,G.;等。人参皂苷Rg1通过糖皮质激素受体相关核因子-κB通路对酒精性肝炎的抗炎功能。民族药理学杂志 2015, 173, 231-240.

评论5获得的结果是否与人体疗法有某种关系? 

回应是的。目前,临床治疗类固醇耐药性哮喘的重要途径是联合使用甾体药物[1,2],因此,联合治疗模式具有巨大的应用潜力。然而,长期服用甾体类药物往往会导致患者产生类固醇耐药性,导致疗效降低甚至无效。我们的研究结果表明,MCG具有缓解类固醇耐药性的潜力,并且长期使用MCG作为天然产品不会产生不利影响。当与甾体类药物联合使用时,MCG可以帮助减少类固醇的剂量,达到降低毒性和增强疗效的效果。

引用

  1. Wang G, Wang F., Gibson P.G, 等.中国严重和不受控制的哮喘——来自澳大利亚严重哮喘网络的横断面调查 J. jThorac Dis, 2017,9(5):1333-1344.
  2. Gibson PG、Yang l.A.、Upham JW 等人。阿奇霉素对持续性不受控制的哮喘成人哮喘发作和生活质量的影响(AMAZES):一项随机、双盲、安慰剂对照试验[柳叶刀,2017,390(10095):659-668。

Reviewer 2 Report

Comments and Suggestions for Authors

With interest, I read the manuscript ijms-3166950. Interesting manuscript based on a data-rich study.

I have minor comments only:

1.        I understand that CSRAM was modeled through a combination of OVA and cold, tight? Perhaps I omitted something, but how was SRAM modeled?

2.        What was done was a murine (allergic) airway inflammation model mimicking human asthma. So it should be given throughout the manuscript.

3.        BAL neutrophil counts were provided. What about eosinophil counts? How can we be sure they were not increased as well, reflecting the airway inflammation of a mixed type? Please, provide the data on BAL eosinophils as well (could be simple cytospin differential staining).

4.        The effects of OVA-based allergic airway inflammation model on (gut) microbiome have already been reported (PMID: 36458896). Please, relate to your data.

5.        In many figures details are much too small to read. Please, amend it.

6.        All abbreviations used in figures and/or their legends must be explained in the legends.

7.        Please, makes sure that the names of the genes are correct and species-specific (https://www.ncbi.nlm.nih.gov/gene), and always written in italics.

Comments on the Quality of English Language

Moderate amendments.

Author Response

我们衷心感谢各位提出的建设性意见。根据评论,对手稿进行了仔细修订。

评论1: 我知道 CSRAM 是通过 OVA 和 cold 的组合建模的,对吧?也许我省略了一些东西,但是SRAM是如何建模的?
回应如您所说,冷刺激类固醇耐药性哮喘模型 (CSRAM) 由 OVA、不可分型流感嗜血杆菌 (NTHi) 和冷刺激 (10 ± 2°C) 建模,而类固醇耐药性哮喘模型 (SRAM) 由 OVA 和 NTHi 建模。两组之间的唯一区别是,CSRAM组从第21天到第31天每天接受6小时的冷刺激,而SRAM组则没有。您怀疑的原因可能是我们没有澄清图表或图例中使用的缩写,而手稿中的图 1A 没有得到很好的呈现。在修订版中,我们澄清了所有缩写,并重新绘制了实验程序图。

评论2: 所做的是模仿人类哮喘的小鼠(过敏性)气道炎症模型。所以它应该贯穿整个手稿。
回应:非常感谢您的建议。我们在整份手稿中都指出,在涉及模型的地方,我们所做的是模仿人类哮喘(过敏性)气道炎症模型。

评论3: 提供了 BAL 中性粒细胞计数。嗜酸性粒细胞计数如何?我们怎么能确定它们没有增加,反映了混合型的气道炎症?请同时提供有关 BAL 嗜酸性粒细胞的数据(可能是简单的细胞离心差异染色)。

回应:首先,我们课题组先前发表的文章结果如下图所示,与OVA诱导的哮喘模型相比,OVA联合不可分型流感嗜血杆菌(NTHi)感染诱导的类固醇耐药哮喘小鼠模型中,中性粒细胞数量显著增加,嗜酸性粒细胞数量显著减少[1]。

其次,其他文献也表明,使用OVA联合流感嗜血杆菌感染诱发的类固醇耐药性哮喘的主要特征是中性粒细胞计数显著增加,嗜酸性粒细胞计数显著减少[2-4]。

第三,在我们的实验前研究中,使用Diff快速染色试剂盒(Solarbio,北京,中国)来检测我们估计的模型中中性粒细胞和嗜酸性粒细胞的计数。CSRA模型组的典型结果显示NTHi感染后中性粒细胞数量显著增加,嗜酸性粒细胞数量显著减少。因此,可以肯定的是,在我们的估计废除模型中,嗜酸性粒细胞计数没有增加,而中性粒细胞计数增加。

最后,我们手稿的H&E染色结果(图4A)也能够证明,与AM或CAM组相比,SRAM或CSRAM组的中性粒细胞数量和比例显着更高,嗜酸性粒细胞的数量和比例显着降低。因此,在我们的正式实验中,使用流式细胞术仅检测中性粒细胞的数量和比例,而使用流式细胞术或Diff快速染色试剂盒未检测到嗜酸性粒细胞的数量。

综上所述,消除冷刺激类固醇耐药的哮喘模型主要是中性粒细胞介导的气道炎症,而不是混合性气道炎症。

引用

  1. 王,G.;庞,Z.;Chen-Yu Hsu, A.;关,X.;冉,N.;袁咏仪;王志强;郭英;郑,R.;Wang, F. SB203580和地塞米松联合治疗可抑制小鼠过敏性哮喘中不可分型的流感嗜血杆菌诱导的 Th17 炎症反应。欧洲药理学杂志 2019, 862.
  2. 2. 埃西尔菲,A.-T.;辛普森,J.L.;邓克利,ML;摩根,L.C.;奥利弗,BG;吉布森,PG;福斯特,PS;Hansbro, P.M. 结合流感嗜血杆菌呼吸道感染和过敏性气道疾病驱动慢性感染和中性粒细胞哮喘的特征。胸部 2012, 67, 588-599.
  3. 3. 徐洋;王轶杰;赵胜涛;王冉;在过敏性气道疾病期间长期暴露于低剂量流感嗜血杆菌可驱动类固醇耐药性中性粒细胞炎症并促进气道重塑。肿瘤靶点 2018, 9, 24898-24913。
  4. 4. 菲尔波特,DJ;埃西尔菲,A.-T.;辛普森,J.L.;霍瓦特,JC;普雷斯顿,J.A.;邓克利,ML;福斯特,PS;吉布森,PG;Hansbro, P.M. 流感嗜血杆菌感染驱动 IL-17 介导的中性粒细胞过敏性气道疾病。PLoS 病原体 2011, 7.

评论4: 基于OVA的过敏性气道炎症模型对(肠道)微生物组的影响已经报道(PMID:36458896)。请与您的数据相关。

回应:非常感谢您的建议。我们引用了上述文章(PMID:36458896)的发现与我们的数据进行比较,并在讨论部分将两者联系起来。

 评论5:在许多数字中,细节太小而难以阅读。请修改它。
回应:我们在修订版本中重新排列了图 1图 3图 6,以最大限度地增加细节量。由于空间限制,其他图形一直保持其原始布局,但我们已上传所有图形的矢量图像,以确保它们可以放大以查看所有细节。

说明6:图表和/或其图例中使用的所有缩写必须在图例中加以解释。

答复:在修订后的文本中,我们在“缩写部分增加了图表和/或图例中包含的NC、CSC、AM、CAM、SRAM、CSRAM、MCGL、MCGM和MCGH的解释,图表和/或图例中所有缩写的解释都在“缩写”部分。

注释 7:请确保基因名称正确且具有物种特异性 (https://www.ncbi.nlm.nih.gov/gene),并始终以斜体书写。
回应:在修订版中,我们确定所有基因都正确命名且具有物种特异性(Mus musculus)。手稿中的所有基因均已改为斜体字。
